# General Anesthesia Compared to Spinal Anesthesia for Patients Undergoing Lumbar Vertebral Surgery: A Meta-Analysis of Randomized Controlled Trials

**DOI:** 10.3390/jcm10010102

**Published:** 2020-12-30

**Authors:** Alessandro De Cassai, Federico Geraldini, Annalisa Boscolo, Laura Pasin, Tommaso Pettenuzzo, Paolo Persona, Marina Munari, Paolo Navalesi

**Affiliations:** 1UOC Anaesthesia and Intensive Care Unit, University Hospital of Padua, 35121 Padua, Italy; federico.geraldini@gmail.com (F.G.); annalisa.boscolo@gmail.com (A.B.); laurapasin1704@gmail.com (L.P.); tompetty86@gmail.com (T.P.); ppersona75@gmail.com (P.P.); marina.munari@aopd.veneto.it (M.M.); pnavalesi@gmail.com (P.N.); 2UOC Anaesthesia and Intensive Care Unit, Department of Medicine-DIMED, University of Padua, 35121 Padua, Italy

**Keywords:** general anesthesia, spinal anesthesia, meta-analysis, vertebral surgery

## Abstract

Vertebral lumbar surgery can be performed under both general anesthesia (GA) and spinal anesthesia. A clear benefit from spinal anesthesia (SA) remains unproven. The aim of our meta-analysis was to compare the early analgesic efficacy and recovery after SA and GA in adult patients undergoing vertebral lumbar surgery. A systematic investigation with the following criteria was performed: adult patients undergoing vertebral lumbar surgery (P); single-shot SA (I); GA care with or without wound infiltration (C); analgesic efficacy measured as postoperative pain, intraoperative hypotension, bradycardia, length of surgery, blood loss, postoperative side effects (such as postoperative nausea/vomiting and urinary retention), overall patient and surgeon satisfaction, and length of hospital stay (O); and randomized controlled trials (S). The search was performed in Pubmed, the Cochrane Central Register of Controlled Trials, and Google Scholar up to 1 November 2020. Eleven studies were found upon this search. SA in vertebral lumbar surgery decreases postoperative pain and the analgesic requirement in the post anesthesia care unit. It is associated with a reduced incidence of postoperative nausea and vomiting and a higher patient satisfaction. It has no effect on urinary retention, intraoperative bradycardia, or hypotension. SA should be considered as a viable and efficient anesthetic technique in vertebral lumbar surgery.

## 1. Introduction

Vertebral lumbar surgery can be performed under both general anesthesia (GA) and spinal anesthesia (SA). Each has possible advantages and complications in the perioperative period [1]. In particular, SA does not require airway device placement for intraoperative sedation and analgesia; however, it could be associated with patient discomfort and intraoperative patient movements [2]. Moreover, fear of neuraxial damage caused by either local anesthetic toxicity or direct damage with an associated prolonged hospital length of stay (LOS) may discourage its use [3]. 

According to a systematic review and meta-analysis published in 2016 [4], patients undergoing lumbar spine surgery under SA required less analgesia in post-anesthesia care units and had less nausea and vomiting (PONV) in the first postoperative day, but no difference in intraoperative hypotension, bradycardia, blood loss, and surgical time was reported.

Given the above, a clear benefit from SA during lumbar spine surgery remains unproven. Furthermore, relevant clinical outcomes remain unexplored.

The primary aim of our meta-analysis is to compare the analgesic efficacy of SA and GA for postoperative pain control in adult patients undergoing lumbar vertebral surgery.

The secondary aim is to evaluate differences in surgery length, perioperative complications (blood loss, hypotension, bradycardia, urinary retention, PONV), need for early analgesics, LOS, patient and surgeon satisfaction, and hospital LOS between the two techniques.

## 2. Materials and Methods

We followed PRISMA (Preferred Reporting Items for Systematic reviews and Meta-Analysis) Statement Guidelines to prepare this manuscript [5].

A review protocol was written before conducting this study and registered in Open Science Framework (reference: KXP3C) on 3 July 2020 [6].

### 2.1. Search Strategy

We performed a systematic research of the medical literature for the identification, screening, and inclusion of articles. The search was performed in the following databases from inception until 6 July 2020: Pubmed, the Cochrane Central Register of Controlled Trials (CENTRAL), and Google Scholar. We also checked the reference lists of included studies. For specific information regarding our search strategy, see Appendix A. We did not apply any restriction on publication type, language, status, and year of publication.

### 2.2. Study Selection

Two researchers (ADC and FG) independently screened titles and abstracts of the identified papers in order to select relevant and not-relevant papers. Each citation was reviewed with full-text retrieval of any citation considered potentially relevant. The following PICOS criteria were used for study inclusion: adult patients (≥18 years old) undergoing vertebral lumbar surgery (P); single-shot SA (I); GA care with or without wounds infiltration (C); analgesic efficacy measured as postoperative pain at six hours (visual analogue scale (VAS) and/or postoperative analgesic requirement at six hours), intraoperative hypotension, bradycardia, length of surgery, blood loss, postoperative side effects (such as PONV and urinary retention), overall patient and surgeon satisfaction, and LOS (O); and randomized controlled trials (RCTs) (S).

### 2.3. Data Extraction and Data Retrieval

After identifying those studies meeting inclusion criteria, two members of our team (FG and ADC) independently reviewed and assessed each of the included studies. Any disagreement on both study selection and data extraction was planned to be solved by a third author (MM) or by contacting the corresponding author. The following information was collected: first author, year of the study, total number of patients per group, postoperative pain at six hours or, if not available, the nearest value within four hours (VAS, numeric rating scale or analogues), postoperative analgesic requirement (% of patients), intraoperative blood loss (mL), surgery length (min), occurrence of intraoperative hypotension (% of patients), bradycardia (% of patients), postoperative PONV (% of patients), urinary retention (% of patients), patient and surgeon satisfaction (as both % and scale), and LOS (days).

Moreover, we extracted the following data in order to perform post-hoc analyses: pain at 24 h, preoperative pain, and wound infiltration.

If data were missing, a request was sent by e-mail to the corresponding author of the study. If no response was received after our initial request, a second request was sent five days later. A third and last request was sent one week after the second one.

### 2.4. Quality Assessment and Quality of Evidence Assessment

Two researchers (ADC and FG) independently evaluated the quality of included RCTs by using the Risk of Bias (RoB) 2 Tool [7]. Disagreements were resolved by discussion with a third researcher (MM).

RoB 2 Tool assesses study quality and risk of bias by exploring five domains (bias arising from the randomization process, bias due to deviations from intended interventions, bias due to missing outcome data, bias in the measurement of the outcome, bias in the selection of the reported result, and each domain is judged on a three-grade scale (low, high, or some concerns). An overall risk of bias is expressed based on the above domains on a three-grade scale (low, high, or some concerns).

We used the Grades of Recommendation, Assessment, Development, and Evaluation (GRADE) approach to assess the quality of evidence related to each of the key outcomes [8]. Starting from “High quality” of evidence, it was downgraded by one level for serious, or by two levels for very serious study limitations, such as risk of bias, indirectness of evidence, inconsistency, imprecision of effect estimates, or potential publication bias.

’Indirectness of evidence’ was considered when subjects, intervention, or outcome were different from those of primary interest for the meta-analysis.

We assessed ‘inconsistency of the outcome’ as follows: (i) confidence intervals not overlapping; (ii) a *p*-value < 0.1 and I^2^ > 50% considering heterogeneity; (iii) or when important differences between studies or subgroups remained without explanation.

‘Imprecision of effect’ occurred in case of small sample size, number of events, and uncertainty about magnitude of effect given by large intervals of confidence.

A potential ‘publication bias’ was recorded when potential bias was detected in the funnel plot.

### 2.5. Statistical Methods

Meta-analysis of data was performed using RevMan version 5.3 (Foundation for Statistical Computing, Vienna, Austria).

The treatment effect for continuous outcomes was expressed as standardized mean difference (SMD) with 95% confidence interval (CI), when the outcome was expressed with different measurement techniques, or mean difference (MD) with 95% CI, when the outcome was derived from the same measurement technique. The treatment effect for dichotomous outcomes was expressed as odds ratio (OR) with 95% CI. Where necessary, we converted reported median and interquartile range or first-third quartile to estimated mean and standard deviation (SD) using Hozo’s method. A random effect model was preferred when I^2^ > 25%.

For assessment of study heterogeneity, the Chi-squared test and I^2^-statistic were used (considering I^2^ values as follows: low: <25%, moderate: 25% to 50%, or high: >50%) [9]. Values of *p* < 0.05 were considered to be statistically significant in all cases.

A predefined subgroup analysis was planned according to surgery (discectomy or laminectomy).

A post-hoc subgroup analysis was planned according to preoperative use of opioids, pain at 24 h, and wound infiltration.

Once the initial statistical analysis was performed, further sensitivity analyses were performed by sequentially removing data from those studies with a high risk of bias and analyzing with a random effect analysis those studies with low heterogeneity.

Zero events were treated by applying a continuity correction, which added one half to each value.

A pre-specified Trial Sequential Analysis (TSA) was performed on each outcome. We estimated the required information size on the calculated minimal intervention effect, considering a type I error of 5% and a power of 80%. This post hoc conservative approach allows us to assess if the data provide convincing evidence of the true effect [10].

## 3. Results

### 3.1. Study Selection and Data Retrieval

Bibliographic search results are shown in the Preferred Reporting Items for Systematic Reviews and Meta-Analyses (PRISMA) diagram (Figure 1). Notably, three RCTs did not report quantitative data. Two papers were excluded because, despite our best efforts, we were not able to retrieve the full text [11,12], in one case [13] the paper did not contain any variable of interest and the authors were not able to provide any missing information.

Eleven studies counting a total of 896 patients entered the quantitative and qualitative analysis [14,15,16,17,18,19,20,21,22,23,24]. All controversies were solved by discussion and the third reviewer was not required.

We asked all the corresponding authors for missing data, and five of them replied to our query. Only two of them, nevertheless, provided part of the missing data required [16,17,18,19,20,21,22,23,24].

Additional records were identified by checking the reference lists of included studies.

### 3.2. Study Characteristics

Among the 896 patients, half (449; 50.12%) underwent GA, while the remainder underwent SA (447; 49.88%). The characteristics of included studies are shown in Table 1 and Appendix A.

There were concerns of bias in ten studies, where one study was evaluated at high risk of bias [23] (Figure 2). Details on risk of bias assessment are available as Appendix A.

## 4. Outcomes

### 4.1. Primary Outcome: Postsurgical Pain

For this outcome, we retrieved data of interest in six studies. [15,16,17,20,21,22]. Two studies evaluated pain at the sixth hour [15,16,17,18,19,20], one between the fourth and the eighth hour [16], one between the second and the third hour [22], and in two studies, pain was described as peak pain [17,21]. All studies used the VAS to assess postoperative pain. However, two studies used a 101-point scale [17,21], while the others used an 11-point scale.

As shown in picture (Figure 3), patients receiving SA reported less pain when compared to the GA group (SMD: −2.32, 95% CI: −3.91 to −0.73, *p*: 0.004, I^2^: 98%).

In the TSA (Figure 4), the cumulative z-score crossed the monitoring boundary for benefit at the first trial yielding an effect that is both statistically and clinically significant. However, the quality of evidence was very low (Appendix A).

Notably, in a post-hoc analysis evaluating pain at 24 h after surgery, there were no differences among the groups (SMD: −0.33, 95% CI: −0.69 to 0.04, *p*: 0.08, I^2^: 60%) [15,20,21,22] (Appendix A).

Only two studies reported the pain before the surgery [22,23], for this reason, we were not able to perform any further analysis.

### 4.2. Secondary Outcomes

All the secondary outcomes analysis discussed below are available as supplementary material for both forest plots (see Supplemental Digital Content 5) and TSA (see Appendix A).

(1)Need for early postoperative analgesics

Six studies reported the need for postoperative analgesics [14,17,18,19,21,23]. All studies referred to the analgesic administration in the post-anesthesia care unit, except for Attari’s paper [14] (see Appendix A).

The overall analysis of these data shows a clear benefit for SA (OR: 11.52, 95% CI:5.12 to 25.93, *p* < 0.001, I^2^ 57%) with a number needed to treat (NNT) of 2 (95% CI 1.9 to 2.6) meaning that one out of every two patients will benefit from the treatment. In the TSA, the cumulative z-score line crossed the monitoring boundary for SA benefit at the first trial. From the second trial, the cumulative z-curve also reached the required information size. We assigned a low quality of evidence to this outcome (see Appendix A).

(2)Blood loss

Blood loss was evaluated in six studies [14,17,18,19,21,23]. Meta-analysis overall effect is in favor of SA (MD: −53.88 mL, 95% CI: −98.13 to −9.63, *p*: 0.02, I^2^: 97%). In the TSA, the cumulative z-curve lies immediately under the monitoring boundary for benefit, not confirming the statistical significance. Given the serious inconsistency of the outcome, we evaluated the quality of evidence as low (see Appendix A).

(3)Surgery length

Almost all the studies evaluated differences in surgery length [14,15,16,17,18,19,21,22,23,24] with mixed results. Both the overall effect and the TSA showed that surgery length is not significantly influenced by the technique (MD: −4.56 min, 95% CI: −13.16 to 4.04, *p*: 0.30, I^2^: 98%). Notably, the standard deviation was not provided in one study [15] and the corresponding author was not able to retrieve it. For this reason, it was excluded in the final analysis. The quality of evidence was very low (see Appendix A).

(4)Intraoperative hypotension and bradycardia

Hypotension and bradycardia were heterogeneously defined among the studies by either a predetermined cut-off value [14,18] or by a decrease from the baseline [17,18,19,21]. Two studies did not define hypotension and bradycardia [15,20]. The intraoperative incidence of bradycardia (OR: 0.74, 95% CI: 0.30 to 1.80, *p*: 0.51, I^2^: 55%) and hypotension (OR: 0.51, 95% CI: 0.23 to 1.11, *p*: 0.09, I^2^: 61%) were not influenced by the anesthesia.

Remarkably, both TSAs showed the z-cumulative line was in close proximity to futility boundaries. We assigned a low quality to both outcomes (see Appendix A).

(5)Nausea and vomiting

Ten studies evaluated postoperative PONV [14,15,16,17,18,19,20,21,22,24]. Patients undergoing GA were more likely to experience PONV (OR: 2.69 95% CI: 0.73 to 4.20, *p* < 0.001, I^2^: 24%) with a NNT of 9.3 (95%CI 6.1 to 15.3).

The TSA z-cumulative line reached the monitoring boundary for benefit at the second trial and the required sample size at the fifth trial. Given the above, we considered the quality of the evidence for this outcome as moderate (see Appendix A).

(6)Urinary retention

Urinary retention was a secondary end-point of seven studies [15,16,17,20,21,22,24]. There was no effect in the overall analysis (OR: 1.15, 95% CI: 0.68 to 1.94, I^2^:0%) with a moderate quality of evidence (see Appendix A).

(7)Length of stay

Results from seven studies [16,17,18,21,22,23,24] showed a lower LOS in patients receiving SA (MD: −0.31 days, 95% CI: −0.41 to −0.21, *p* < 0.001, I^2^: 54%). This result was confirmed by the TSA with the cumulative z-score reaching the required sample size at the second trial. We assigned a low quality of evidence to this outcome (see Appendix A).

(8)Patient and surgeon satisfaction

We were able to extract data from six studies regarding patient satisfaction (%) [14,15,20,21,22,23] and from three studies regarding surgeon satisfaction [18,19,21].

While patients tend to prefer SA (OR: 0.38, 95% CI: 0.12 to 1.16, *p* < 0.001, I^2^ 37%), surgeons preferred GA (MD: −11.08, 95% CI: −13.56 to −8.60, *p* < 0.001, I^2^: 56%). Both results were confirmed in the TSA, but the quality of evidence was moderate and low, respectively (see Appendix A).

### 4.3. Publication Bias

Notwithstanding the lack of clear asymmetry at visual inspection, a definite interpretation of the funnel plots was not possible due to the paucity of studies (see Appendix A).

### 4.4. Sensitivity Analysis

Sensitivity analysis comparisons are shown in Table 2.

Removal of the Kilic et al. [23] study, which was judged to be at high risk of bias, did not change the overall effect for the following outcomes: analgesic requirement, LOS, patient satisfaction, blood loss, and surgery length.

However, we identified this study as the source of heterogeneity for the analgesic requirement outcome (I ^2^ dropped from 57% to 0%).

Estimate results from both random and fixed effect models were extremely similar and did not modify the statistical analysis results.

### 4.5. Subgroup Analysis

Studies were sub-grouped in studies evaluating laminectomy or discectomy. Studies evaluating both surgeries and studies without a clear description of the surgery were excluded from the subgroup analysis. Results were similar to the overall analysis without a clear advantage based on the type of surgery.

Only two studies evaluated preoperative pain improvement over baseline and no study investigated both opioid naive compared to opioid dependent subgroups and infiltration of the wound. For this reason, it was not possible to perform the above subgroup analysis.

## 5. Discussions

Comparison between SA and GA is still a hot topic in literature. Two previous meta-analyses [3,4] had already performed a valid comparison between these two methodologies during spine surgery. Specifically, Meng et al. [4] conducted an accurate data extraction in 2016, but they did not investigate patient and surgeon satisfaction.

However, the Committee on Quality of Health Care in 2001 defined a patient-centered health system as a healthcare quality goal [25]. We may assume that satisfaction of the patient is a key-component of patient-oriented healthcare. Although both surgeon and patient point towards a positive outcome of the intervention, the goals may be different. In fact, an elevated patient satisfaction does not necessarily correspond to safe and effective care [26]. Given the above, both patient and surgeon satisfaction should be two important goals in medicine, representing an index of safe, effective, and patient-oriented healthcare. In addition, this meta-analysis is four years old [4]. Although there is not a consensus regarding time to meta-analysis update [27], an analysis shows that 23% of reviews are out of date within two years of publication, with a median time to require an update of 5.5 years [28]. About the second study, Zorrilla-Vaca et al. [3] conducted a nice systematic review of the literature but SA was not considered as an outcome. Indeed, the increased interest about patient and surgeon satisfaction, time lapsed from Meng’s paper [4], missed focus on spinal anesthesia in Zorrilla-Vaca’s manuscript [3], the possibility to assess bias using the recently developed RoB2 tool [7], the plan to carry out both sensitivity and subgroup analysis, the use of trial sequential analysis (TSA) [29], and the GRADE tool drove us to conduct this meta-analysis.

Our findings suggest that SA decreases early postoperative pain after vertebral surgery. Patients undergoing vertebral surgery experience severe acute pain that can last four days [30].

An effective strategy for pain relief is essential to facilitate early mobilization and reduce hospital LOS [31].

Our study shows that SA lowers postoperative pain at the sixth hour and the need for analgesic in the PACU. The preemptive effect of SA abolishing sensitization through the nociceptive pathway may explain this result [14].

Actually, interfascial blocks such as the erector spinae plane (ESP) block and retrolaminar block drew attention in lumbar vertebral surgery because they are able to provide pain relief through multiple pathways [32,33]. A future analysis, incorporating these techniques, could be of paramount importance to define the best multimodal analgesic strategy for this surgery.

PONV etiology is complex and multifactorial [34]. Both SA and GA have risk factors for this postoperative complication. In particular, inhalational agents and opioids are strongly associated with PONV after GA, while sympathetic nervous system blockade caused by SA can cause severe PONV. However, it is already known that PONV is 9 times less frequent among patients receiving SA than those receiving GA [35]. Our meta-analysis confirms this result, and there may be a direct consequence of a reduced opioid consumption in the perioperative period.

Therefore, the patients’ preference of SA is not surprising. Nonetheless, under SA, surgery times are forced by local anesthetic pharmacokinetics. Moreover, patients’ movements could interfere with surgery, making it less comfortable for the surgeon. The urge to finish the surgery, patients’ movements, and the impossibility to check neurologic status immediately after the operation may explain the preference of the surgeon for GA. However, no study evaluated the patient’s movement with surgery failure or postoperative neurologic damage, leaving space for future research.

Intraoperative and postoperative complications did not differ among the groups. Even if blood loss resulted significantly lower, it is debatable if a difference of 50 mL is clinically relevant. Moreover, considering that LOS was slightly shorter in the SA group, the fear of severe complications causing prolonged hospital stay related to SA [3] is not justified.

### Limitations

Our study has limitations deserving discussion. First of all, although all the included studies were RCTs with similar key characteristics (methodology and main outcomes), the study heterogeneity was high for some outcomes. However, the use of a random effect meta-analysis for the outcome with a moderate and high heterogeneity and of a sensitivity analysis for the low heterogeneity outcomes add strength to our conclusions.

Second, we limited our search to three databases; however, we recognize that other literature databases exist and this could have led to us missing some papers. Nonetheless, we used a rigorous systematic review and meta-analytic methods including a reproducible and comprehensive literature search strategy, clearly defined inclusion criteria, and duplicate citation review, data extraction, and quality assessment. Moreover, a protocol for our systematic review was pre-published.

Third, some outcomes were evaluated using SMD assuming that the differences in standard deviations among studies reflect differences in measurement methods and not real differences in variability.

Fourth, the heterogeneity in local anesthetics, different GA protocols, and heterogenous cutoff for some outcomes limit our conclusions.

Fifth, several secondary outcomes were analyzed and reported. Indeed this increases the possibility of a multiplicity issue and potentially leads to an increase in the overall type I error rate for all outcomes in totality.

In conclusion, SA in vertebral lumbar surgery decreases postoperative pain and analgesic requirement in the PACU. It is associated with a reduced incidence of PONV and a higher patient satisfaction. It has no effect on urinary retention, intraoperative bradycardia, or hypotension.

Given the above, SA should be considered as a viable and efficient anesthetic technique in vertebral lumbar surgery.

## Figures and Tables

**Figure 1 jcm-10-00102-f001:**
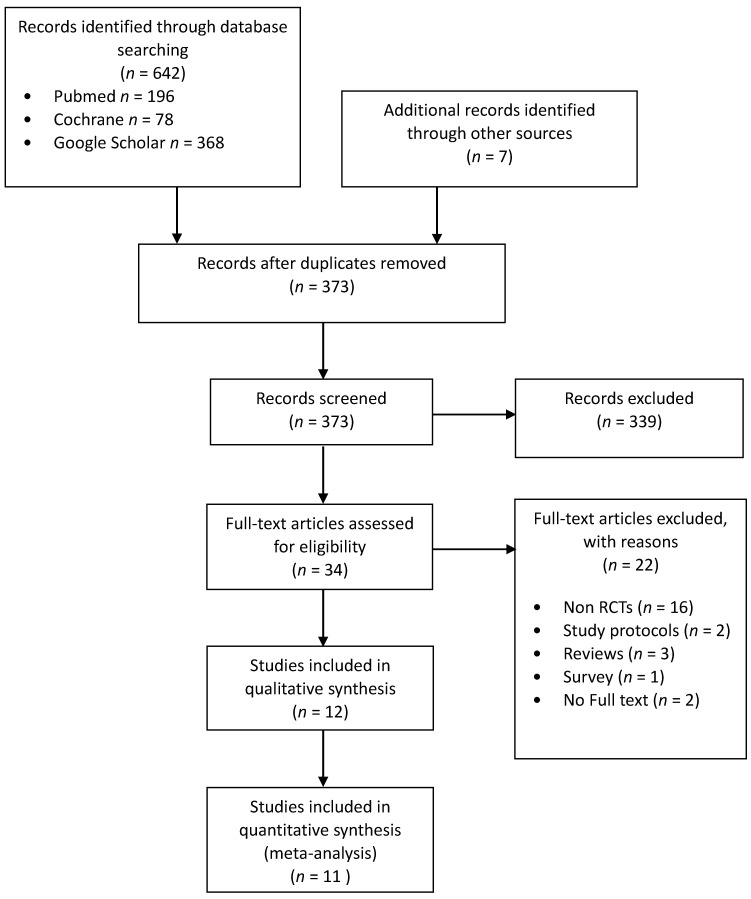
The PRISMA flowchart.

**Figure 2 jcm-10-00102-f002:**
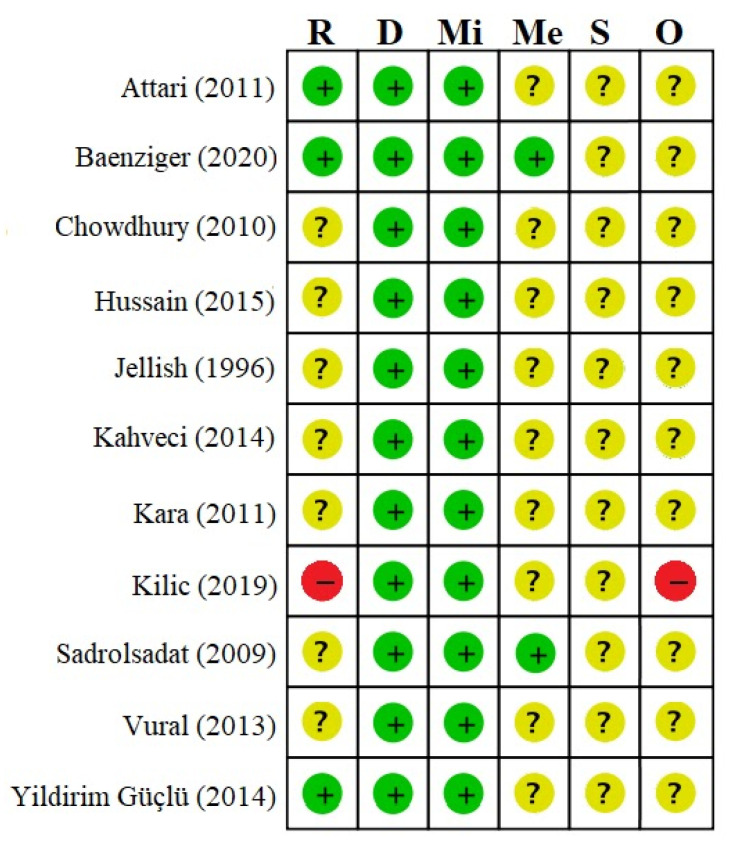
Summary of bias evaluated by the Risk of Bias 2 Tool. R: Bias arising from the randomization process, D: Bias due to deviations from intended interventions, Mi: Bias due to missing outcome data, Me: Bias in measurement of the outcome, S: Bias in the selection of the reported result, and O: Overall risk of bias. Green “+”: Low risk of bias, Yellow “?”: Some concerns, Red “-”: High risk of bias.

**Figure 3 jcm-10-00102-f003:**
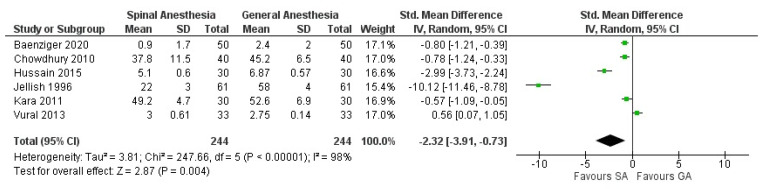
Forest plot of postoperative pain.

**Figure 4 jcm-10-00102-f004:**
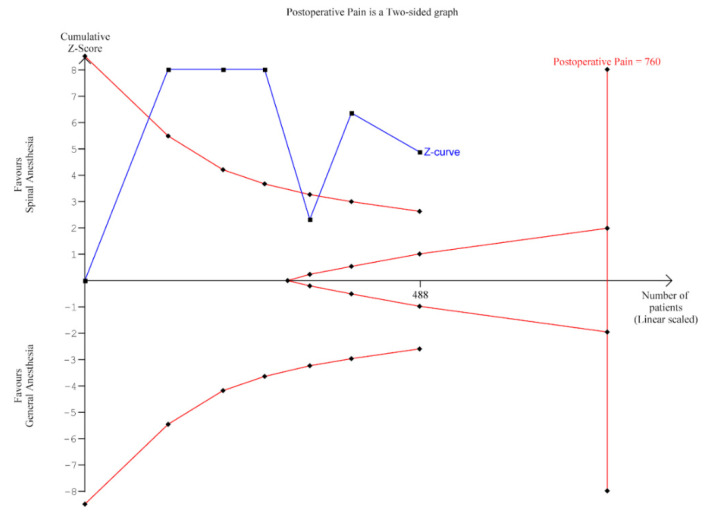
Trial Sequential Analysis of postoperative pain.

**Table 1 jcm-10-00102-t001:** Study characteristics.

Study	N (% F)	Inclusion Criteria			Protocols			Surgery	Time to Pain Assessment
		Age	ASA-PS	BMI (Kg/m^2^)	SA	GA	PO Therapy		
Attari (2011) [14]	72 (46%)	18–60	I–II	NR	3.0–3.2 mL Hyperbaric Bupivacaine 0.5% + 25 mcg Fentanyl	- Induction: propofol, lidocaine, fentanyl, atracurium - Maintenance: isoflurane 1,2%, N_2_0 50%	Pethidine 0.4 mg/kg on VAS (rescue pethidine 0.2 mg/kg).	Laminectomy, Discectomy	NR
Baenziger (2020) [22]	100 (46%)	Adult	I–III	NR	3.0–4.0 mL Hyperbaric Bupivacaine 0.5% + 25 mcg Fentanyl	- Induction: propofol, fentanyl, atracurium - Maintenance: Propofol TCI, Remifentanil TCI	NR	Laminectomy, Discectomy	3 h
Chowdhury (2010) [15]	80 (38%)	Adult	I–II	NR	2.5–2.8 mL Hyperbaric Bupivacaine 0.5% + 12.5 mcg Fentanyl	- Induction: propofol, fentanyl, rocuronium - Maintenance: halothane 0.8%, N_2_0 60%	Pethedine 2 mg/kg six hourly and on request.	Discectomy	6 h
Hussain (2015) [16]	60 (50%)	20–50	I–II	NR	2 mL Bupivacaine 0.75%	- Induction: propofol, atracurium - Maintenance: sevoflurane 1.5–2%, nalbuphine	NR	Micro- discectomy	Peak at 6 h
Jellish (1996) [17]	122 (46%)	Adult	I–III	NR	1.5 mL Hyperbaric Bupivacaine 0.75%	- Induction: thiopental, fentanyl, vecuronium - Maintenance: isoflurane, N_2_O 70%	PACU: morphine 2 mg IV ward:meperidine 25–50 mg IV or 50–100 mg intramuscularly.	Laminectomy, Discectomy	Peak
Kahveci (2014) [18]	80 (38%)	≥18	I–II	≤25	3 mL Hyperbaric Bupivacaine 0.5%	- Induction: propofol, fentanyl, atracurium - Maintenance: sevoflurane 1.5–2%, atracurium	Pethedine 25 mg IV on VAS.	Single-level spinal surgery	NR
Kara (2011) [21]	60 (45%)	Adult	I–II	NR	2 mL Levobupivacaine 0.5%	- Induction: propofol, fentanyl, rocuronium - Maintenance: desflurane 6%, N_2_O 40–60%	Morphine 2 mg on VAS.	Discectomy	Peak
Kilic (2019) [23]	111 (45%)	18–65	I–III	NR	3 mL Hyperbaric Bupivacaine 0.5%	- Induction: propofol, fentanyl, rocuronium - Maintenance: sevoflurane 1.5–2%, remifentanil	NR	Micro- discectomy	3 h
Sadrolsadat (2009) [19]	100 (-)	Adult	I–III	NR	4 mL Bupivacaine 0.5%	- Induction: propofol, fentanyl, atracurium - Maintenance: propofol, alfentanil, atracurium	Pethedine 25 mg IV on VAS (lock 30 min in PACU and 4 h in ward).	Laminectomy	NR
Vural (2014) [20]	66 (-)	23–74	ND	NR	4 mL Hyperbaric Bupivacaine 0.5%	- Induction: thiopental, fentanyl, rocuronium - Maintenance: desflurane 5–6%, N_2_O 40–60%,fentanyl	NR	Disc herniation surgery	6 h
Yildirim Güçlü (2014) [24]	56 (-)	18–60	I–II	≤35	3 mL Hyperbaric Bupivacaine 0.5%	- Induction: thiopental, fentanyl, vecuronium - Maintenance: desflurane 4–5%, N_2_O 50%, remifentanil	Pethidine 0.5 mg/kg on VAS (Rescue pethidine 0.2 mg/kg).	Micro- discectomy	NR

ASA-PS: ASA Physical Status, F: females, GA: General Anesthesia, SA: Spinal Anesthesia, PACU: Post-Anesthesia Care Unit, PO: Post-Operative, NR: Not Reported.

**Table 2 jcm-10-00102-t002:** Sensitivity Analysis.

	Total Effect (95% CI)	I^2^	*p*-Value	Total Effect (95% CI)	I^2^	*p*-Value
High risk of bias						
	Included			Excluded		
Analgesic requirement	OR 11.52 (5.12 to 25.93)	57%	<0.001	OR 8.31 (5.05 to 13.70)	0%	<0.001
LOS	MD −0.31 (−0.41 to −0.21)	54%	<0.001	MD −0.28 (−0.37 to −0.18)	40%	<0.001
Blood loss (mL)	MD −53.88 (−98.13 to −9.63)	97%	0.02	MD −51.10 (−102.14 to −0.06)	98%	0.05
Patient satisfaction	OR 0.38 (0.12 to 1.163)	37%	0.09	OR 0.39 (0.10 to 1.43)	49%	0.15
Surgery length	MD −4.56 (−12.16 to 4.04)	98%	0.30	MD −2.68 (−11.69 to 6.33)	98%	0.56
Fixed vs. random effect						
	Fixed effect			Random effect		
Nausea and Vomiting	OR 2.69 (1.73 to 4.20)	24%	<0.001	OR 2.52 (1.43 to 4.44)	24%	0.001

Note: CI: Confidence Intervals, OR: Odds Ratio, MD: Mean Difference.

## Data Availability

The data presented in this study is fully available in the article itself and in its Appendix A.

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
