# Peer review of "General Anesthesia Compared to Spinal Anesthesia for Patients Undergoing Lumbar Vertebral Surgery: A Meta-Analysis of Randomized Controlled Trials"

_jcm, 2020, doi:10.3390/jcm10010102_

Round 1

Reviewer 1 Report

Overall I believe the meta-analysis was done well and I do not have any major concerns on the statistical analysis (the focus of my main review).  My one minor concerns is the number of secondary outcomes analyzed in multiple different sub-groups. Several secondary outcomes were analyzed and reported on with no mention of the possibility of a multiplicity issue nor the potential effect on the overall type I error rate for all outcomes in totality. I think this fact is an additional limitation the authors should mention. 

Beyond above, there were several areas that I had minor comments on.  These are places where the words chosen were a bit awkward, therefore, they are listed along with a suggestion.

Lines 124-126

The wording of this sentence is awkward.  Suggest a change to:

"Once the initial statistical analysis was performed, further sensitivity analyses were performed by sequentially removing data from those studies with a high risk of bias and analyzing with a random effect analysis those studies with low heterogeneity.”

Line 127

Suggest a change to:

“Zero events were treated by applying a continuity correction which added one half to each value.”

Line 131

No statistical test can prove an effect, so I would suggest changing the wording of this sentence.  Potentially to “assess if the data provide convincing evidence of the true effect”

Line 135

I assume this sentence is meant to say that three RCTs did not “report” quantitative data?

Line 146

50.12% is not almost half, it is half.  And 447 out of 896 undergoing SA is actually 49.88%.  I would restate the sentence.

Table 1 lists the study characteristics of the 11 studies included in the meta-analysis.  Of these, some report that actual age of the patients in each study and others report only that they were adults.  For the four studies that did not report age, and have an “NR” listed, was it verified that they were adult patients?  And if so, why is the “NR” listed and not “Adult”?

Line 304-305

There seems to be text referring to journal directions “5. Conclusions This section is not mandatory, but can be added to the manuscript if the discussion is unusually long or complex.”  It should be removed.

Author Response

Q1:Overall I believe the meta-analysis was done well and I do not have any major concerns on the statistical analysis (the focus of my main review).  My one minor concerns is the number of secondary outcomes analyzed in multiple different sub-groups. Several secondary outcomes were analyzed and reported on with no mention of the possibility of a multiplicity issue nor the potential effect on the overall type I error rate for all outcomes in totality. I think this fact is an additional limitation the authors should mention. 

Beyond above, there were several areas that I had minor comments on.  These are places where the words chosen were a bit awkward, therefore, they are listed along with a suggestion.

R1: Dear reviewer, I would like to thank you for your comments. We tried to respond to your concerns and we modified our manuscript as below described. Moreover we added multiplicity issue in limitations section of the manuscript

Q2:Lines 124-126:The wording of this sentence is awkward.  Suggest a change to:

"Once the initial statistical analysis was performed, further sensitivity analyses were performed by sequentially removing data from those studies with a high risk of bias and analyzing with a random effect analysis those studies with low heterogeneity.”

 Line 127:Suggest a change to:

“Zero events were treated by applying a continuity correction which added one half to each value.”

Line 131:No statistical test can prove an effect, so I would suggest changing the wording of this sentence.  Potentially to “assess if the data provide convincing evidence of the true effect”

 R2: We warmly welcome your suggestion. We restate the sentences as you suggested.

Q3: Line 135:I assume this sentence is meant to say that three RCTs did not “report” quantitative data?

 R3: We agree that we were not clear and we modified accordingly.

Q4: Line 146:50.12% is not almost half, it is half.  And 447 out of 896 undergoing SA is actually 49.88%.  I would restate the sentence.

R4:  We are sorry for our mistake. We corrected the percentage. Moreover we agree that 50.12% corresponds to half .

Q5:Table 1 lists the study characteristics of the 11 studies included in the meta-analysis.  Of these, some report that actual age of the patients in each study and others report only that they were adults.  For the four studies that did not report age, and have an “NR” listed, was it verified that they were adult patients?  And if so, why is the “NR” listed and not “Adult”?

R5: All studies included only adult patients, but some studies do not report the age range. To clarify this point we modified NR to Adult.  

Q6:Line 304-305

There seems to be text referring to journal directions “5. Conclusions This section is not mandatory, but can be added to the manuscript if the discussion is unusually long or complex.”  It should be removed.

Q7: We removed the section “conclusions”. However we leave a sentence at the ending of the manuscript to summarize our findings.

Reviewer 2 Report

Good review of what we all know about benefits of spinal anesthesia. Not too surprising no difference in pain after spinal worn off, but good to read. Should publish this.

Author Response

We would like to thank the reviewer for his kind words about our manuscript.

Reviewer 3 Report

In their manuscript, entitled “General Anesthesia Compared to Spinal Anesthesia 2 for Patients Undergoing Lumbar Vertebral Surgery: a 3 Meta-Analysis of Randomized Controlled Trials”, the authors are presenting the results of a meta-analysis regarding a possible benefit of spinal anesthesia in patients undergoing spine surgery. They found a reduction in pain and PONV along with a higher patient satisfaction in the spinal anesthesia group.

The overall idea of the study is interesting and the manuscript is well-written and the description of the Methods is sound. The presentation of the results is adequate and elaborate. The discussion puts the results well into context. The novelty of the current analysis might not be as high, given the fact that another meta-analysis has already been published (by other authors) in 2017 (Reference 4, Meng et al) and almost no new studies have been published after that. However, the authors clearly state, that they now also examined patient and surgeon satisfaction, which is a valid point.

Due to the high quality of the conduction of the analysis and the valid results, I think the current manuscript might be worth to be published.  

There are only very few further issues within this manuscript, which I would kindly like to ask the authors to address before that:

Minor:

  1. Line 17: “wounds” should be changed to “wound”
  2. Figure 1: This figure is of minor quality regarding its resolution. This should be improved.
  3. Lines 303-305: Please remove the spaceholder starting with “5. Conclusions...”.
  4. There are several minor linguistic matters, which a native speaker should be able to solve easily. I therefore recommend the authors to seek linguistic support.

Author Response

  • Line 17: “wounds” should be changed to “wound”

Done

  • Figure 1: This figure is of minor quality regarding its resolution. This should be improved.

A high quality image has been submitted to editorial board in a zip file as requested. If the file submitted is still low in resolution we will submit a version with a higher dpi

  • Lines 303-305: Please remove the spaceholder starting with “5. Conclusions...”.

We removed the "5.conclusion" as requested by reviewer 1

  • There are several minor linguistic matters, which a native speaker should be able to solve easily. I therefore recommend the authors to seek linguistic support

A native english speaker read and corrected some typos. We would like to thank the reviewer for this comment that would lead to an improved quality of our paper